# Biochemical Characterization of an Arabinoside Monophosphate Specific 5′-Nucleotidase-like Enzyme from *Streptomyces antibioticus*

**DOI:** 10.3390/biom14111368

**Published:** 2024-10-27

**Authors:** Yuxue Liu, Xiaobei Liu, Xiaojing Zhang, Xiaoting Tang, Weiwei Su, Zhenyu Wang, Hailei Wang

**Affiliations:** Henan Engineering Research Center of Bioconversion Technology of Functional Microbes, College of Life Science, Henan Normal University, Xinxiang 453007, China; liuxiaobei2023@stu.htu.edu.cn (X.L.); zhangxiaojing20229@stu.htu.edu.cn (X.Z.); tangxiaoting@stu.htu.edu.cn (X.T.); 2404183082@stu.htu.edu.cn (W.S.); wangzhenyu@htu.edu.cn (Z.W.)

**Keywords:** 5′-nucleotidase, phosphohydrolase, arabinoside, *Streptomyces*

## Abstract

To investigate the function of the gene *penF* in the pentostatin and vidarabine (Ara-A) biosynthetic gene cluster in *Streptomyces antibioticus* NRRL 3238, PenF was recombinantly expressed and characterized. Enzymatic characterization of the enzyme demonstrated that PenF exhibited metal-dependent nucleoside 5ʹ-monophosphatase activity, showing a substrate preference for arabinose nucleoside 5ʹ-monophosphate over 2ʹ-deoxyribonucleoside 5ʹ-monophosphate and ribonucleoside 5ʹ-monophosphate. Metal ions such as Mg^2+^ and Mn^2+^ significantly enhanced enzyme activity, whereas Zn^2+^, Cu^2+^, and Ca^2+^ inhibited it. For vidarabine 5′-monophosphate, the *K*_m_ and *k*_cat_ values were determined to be 71.5 μM and 33.9 min^−1^, respectively. The *k*_cat_*/K*_m_ value was 474.1 mM^−1^·min^−1^ for vidarabine 5-monophosphate and was 68-fold higher than that for 2′-deoxyadenosine 5′-monophosphate. Comparative sequence alignment and structural studies suggested that residues outside the primary substrate-binding site are responsible for this substrate specificity. In conclusion, PenF’s activity toward vidarabine 5ʹ-monophosphate likely plays a role in the dephosphorylation of precursors during Ara-A biosynthesis.

## 1. Introduction

5′-Nucleotidases are ubiquitous enzymes present across all domains of life, catalyzing the hydrolytic dephosphorylation of 5′-nucleoside monophosphates to their corresponding nucleosides and inorganic phosphate. These enzymes perform diverse functions depending on their cellular location, whether in vertebrates or microorganisms. In microbes, intracellular 5′-nucleotidases play pivotal roles in regulating cellular metabolism, particularly concerning intracellular nucleotides [1,2]. Several microbial 5′-nucleotidases belong to the histidine-aspartate (HD) domain superfamily, characterized by a tandem HD dyad that coordinates metal ions [3]. Research has demonstrated that certain microbial intracellular 5′-nucleotidases are essential not only in purine and pyrimidine salvage pathways, but also in the biosynthesis of other critical metabolites [3]. For instance, the HD domain superfamily 5′-nucleotidase member YfbR catalyzes the conversion of 2′-deoxycytidine-5′-monophosphate to deoxycytidine, an important step in the deoxycytidine pathway and de novo synthesis of thymidylate in *E. coli* [4]. Similarly, YitU, an HD domain superfamily 5′-nucleotidase from the *Bacillus species*, is involved in riboflavin biosynthesis [5]. These enzymes function as phosphohydrolases and exhibit both phosphomonoesterase and phosphodiesterase activities, as well as displaying broad substrate specificity towards ribonucleoside monophosphates, deoxyribonucleoside monophosphates, nucleoside diphosphate sugars, sugar phosphates, and other molecules. Moreover, some 5′-nucleotidases also participate in the metabolism of nucleotide analogs, such as nucleoside antibiotics [6].

Nucleoside antibiotics represent a diverse class of secondary metabolites with nucleoside structures, recognized for their promising biomedical activities, including antitumor, antiviral, and anticancer properties [7,8]. These natural products are valuable candidates for pharmaceutical development [9]. However, the biosynthetic mechanisms underlying these nucleoside antibiotics remain largely elusive. Over the past two decades, significant progress has been made in elucidating their biosynthetic pathways [10,11,12]. Several gene clusters involved in nucleoside antibiotic biosynthesis have been identified, revealing novel enzymatic mechanisms [11,13]. For example, the *cof* and *for* gene clusters are responsible for the biosynthesis of coformycin and formycin, respectively, in *Streptomyces kaniharaensis* [14,15]. Similarly, the *pen* gene cluster responsible for the biosynthesis of pentostatin (PTN) and arabinofuranosyladenine (Ara-A, vidarabine) has been identified and characterized in *Streptomyces antibioticus* NRRL 3238. Furthermore, based on bioinformatic and enzymatic investigation, the biosynthetic pathways of these nucleoside antibiotics have also been proposed [13,16]. Notably, some biosynthetic pathways partially overlap with those of primary metabolites such as amino acids and nucleosides. For instance, the early biosynthetic steps of coformycin and PTN share similarities with L-histidine biosynthesis [15]. Despite this progress, the precise biological functions of most individual enzymes within these gene clusters remain to be fully experimentally characterized.

As members of nucleoside antibiotics, Ara-A and PTN exhibit a wide spectrum of biological and pharmacological properties, including antibacterial, antitrypanosomal, anticancer, and antiviral [17,18]. Currently, chemical synthesis is the primary method for Ara-A producing, while the fermentation yield of microbial-derived PTN is low and insufficient to meet medical demands [18]. The large-scale production of these nucleoside antibiotics through systematic metabolic engineering will become an inevitable trend, but challenges persist due to the unclear function of related enzymes and elusive biosynthesis pathways. Therefore, characterizing individual enzymes within these gene clusters and elucidating the biosynthesis pathways of PTN and Ara-A in actinomycetes is particularly important.

In the present study, we aimed to investigate the function of the gene *penF* within the pentostatin PTN and Ara-A biosynthetic gene cluster in *Streptomyces antibioticus* NRRL 3238. We recombinantly expressed and characterized PenF, revealing its function as a 5′-nucleotidase with activity towards a range of substrates. The enzyme was found to catalyze the dephosphorylation of ribonucleoside monophosphates, deoxyribonucleoside monophosphates, and arabinosine monophosphate, with the highest catalytic activity observed for vidarabine 5-monophosphate (Ara-AMP) (Figure 1).

## 2. Materials and Methods

### 2.1. Materials and Chemicals

DNA polymerase was purchased from Mei5 Biotechnology (Beijing, China). Substrates, including Ara-AMP, adenosine-2′(3′)-monophosphate (2′(3′)-AMP), 2′-deoxyinosine 5′-monophosphate (dIMP), 2′-deoxyadenosine 5′-monophosphate (dAMP), 2′-deoxycytidine 5′-monophosphate (dCMP), 2′-deoxythymidine 5′-monophosphate (dTMP), 2′-deoxyguanosine 5′-monophosphate (dGMP), inosine 5′-monophosphate (IMP), adenosine 5′-monophosphate (AMP), uridine 5′-monophosphate (UMP), xanthosine 5′-monophosphate (XMP), cytidine 5′-monophosphate (CMP), guanosine 5′-monophosphate (GMP), etc., were obtained from Shanghai Biotech (Shanghai, China) and Shyuanye (Shanghai, China). Additional analytical biological materials and chemical reagents were sourced from local suppliers.

### 2.2. Gene, Strains and Plasmids

The nucleotide sequence of the *penF* gene (NCBI: KJ856912.1) from *Streptomyces antibioticus* NRRL 3238 was obtained from the NCBI database (https://www.ncbi.nlm.nih.gov) and optimized according to the *Escherichia coli* codon preferences. The gene was then synthesized by SynBio Technologies (Suzhou, China). The optimized *penF* gene was cloned into the pET28a expression vector between the *Nde* I and *Xho* I sites, ensuring it was in-frame with a C-terminal his-tag for expression in *E. coli*. The PenF amino acid sequence was identified through sequence comparison in the NCBI database (AKA87335.1).

The bacterial strains and plasmids used in this study are listed in Table 1. *E. coli* strain DH5α was used for plasmid construction, while BL21(DE3) was employed for the overexpression of recombinant proteins.

### 2.3. Site-Directed Mutagenesis

Site-directed mutagenesis was performed using a restriction-free cloning method [19]. Mutagenic primers (Appendix A) for single-site mutations were designed by substituting the target codon. PCR was used to generate nicked plasmids carrying the desired mutations, with the template plasmid pET28a-PenF and the corresponding primers. The resulting PCR products were treated with *Dpn* I at 37 °C and then transformed into *E. coli* DH5α cells. Successful mutations were confirmed by DNA sequencing (Shanghai Biotech, Shanghai, China).

### 2.4. Preparation of Recombinant Protein

The plasmid pET28a harboring either wild-type (WT) or mutant PenF was transformed into BL21(DE3) cells for protein overexpression. Cultures were grown at 37 °C in Luria Bertani medium containing 50 μg/mL kanamycin. When the optical density at 600 nm (OD_600_) reached 0.6, protein expression was induced with 0.2 mM IPTG, followed by incubation at 16 °C 200 rpm for 24 h. Cells were harvested by centrifugation at 8000× *g*, washed. Cell pellets were disrupted by sonication and resuspended in native binding buffer (50 mM NaH_2_PO_4_, 500 mM NaCl, pH 8.0). After cell disruption, the supernatant was obtained by centrifugation at 15,000× *g* for 20 min. Soluble proteins were purified using an Ni-NTA purification kit (Invitrogen) and stored in 50 mM of Tris-HCl (pH 7.5) with 20% (*v*/*v*) glycerol at −80 °C. Protein concentrations were determined using a Bio-Rad protein assay kit, with bovine serum albumin as the standard. Purity was assessed via SDS-PAGE.

### 2.5. MALDI-TOF Assays

The matrix-assisted laser desorption ionization time-of-flight (MALDI-TOF) mass spectra were obtained from Biflex II (Bruker, Germany). The instrument was equipped with a nitrogen laser (λ = 337 nm) to desorb and ionize the samples. The relative laser intensity was used at 90%, the frequency was 1000 Hz, and the quality axis range was 10–150 kDa.

### 2.6. Activity Assays

Nucleotidase activity assays were conducted in a 0.1 mL reaction mixture containing 50 mM HEPES buffer (pH 7.5), 1 mM dAMP, and appropriately diluted enzyme. The reaction was initiated by adding the enzyme solution and allowed to proceed at 37 °C for 10 min before being terminated at 4 °C. Blank controls were performed without the enzyme or substrate. The activity of purified PenF was quantified by measuring the accumulation of deoxyadenosine levels using high-performance liquid chromatography (HPLC). Reaction samples were filtered through a 0.22 μm membrane, and a 20 μL aliquot was injected for HPLC analysis. A C18 column (Agilent C18, 5 μm, 4.6 × 250 mm) was employed, with a mobile phase of methanol/100 mM phosphate buffer (15:85, *v*/*v*) at pH 7.5 and a flow rate of 1 mL/min. UV absorption at 245 nm was used for detection. The characterization of deoxyadenosine and vidarabine (Ara-A) was conducted using liquid chromatography-mass spectrometry (LC-MS). The LC system employed a C18 column (Waters ACQUITY BEH C18, 50 × 2.1 mm, 1.7 μm), with a mobile phase consisting of methanol and 0.1% (*v*/*v*) aqueous formic acid solution (15:85, *v*/*v*), at a flow rate of 0.3 mL/min. The MS system operated in a positive ion mode with key parameters set as follows: the capillary voltage at 3.5 kV and the desolvation temperature at 300 °C. The mass scan range was configured between 60 and 900 *m*/*z*.

Nucleotidase activity assays for mutant proteins were performed in a 0.1 mL reaction mixture comprising 50 mM HEPES buffer at pH 7.5, 1 mM dAMP or Ara-AMP, and 0.2 mg/L purified WT PenF or mutants. The reaction proceeded at 37 °C for 1 min and was terminated at 4 °C. The activity was quantified by measuring the accumulation of deoxyadenosine or Ara-A levels using the HPLC system.

### 2.7. Effect of Metal Ions and pH on Enzyme Activity

To assess the effect of metal ions on enzyme activity, 1 mM of various metal ions (Mg^2+^, Fe^2+^, Mn^2+^, Co^2+^, Cu^2+^, Zn^2+^, Ni^2+^, Ca^2+^) or EDTA was added to reaction mixtures containing 1 mM dAMP in 50 mM HEPES buffer at pH 7.5. The experiment without adding any metals or chelator was performed as a control. The pH dependence of phosphatase activity toward dAMP (1 mM) was evaluated in the presence of 1 mM MgCl_2_ and purified PenF. Assays were conducted in various 50 mM buffer systems: glycine-HCl (pH 2.0–3.0), citrate-NaOH (pH 3.0–6.0), potassium phosphate (pH 6.0–7.5), HEPES (pH 7.0–8.0), Tris-HCl (pH 7.5–9.5), glycine-NaOH (pH 9.5–10.0), and NaHCO_3_-NaOH (pH 10.0–11.0). Enzyme activity was determined by quantifying the accumulation of deoxyadenosine levels using HPLC.

### 2.8. Substrate Range

The substrate specificity of PenF was evaluated using Ara-AMP, 2′(3′)-AMP, dIMP, dAMP, dCMP, dTMP, dGMP, IMP, AMP, UMP, XMP, CMP, and GMP. Enzyme assays were performed in 50 mM HEPES buffer (pH 7.5) supplemented with 1 mM MgCl_2_, 1 mM nucleotide, and 0.3 mg/mL PenF. Reactions were initiated by adding the enzyme solution, proceeded at 37 °C for 10 min, and were terminated at 4 °C. Enzyme activities were determined by quantifying either the decrease in substrate levels or the accumulation of corresponding product nucleoside levels using HPLC.

### 2.9. Kinetic Parameters

Kinetic parameters were determined by measuring enzyme activities at various substrate concentrations, ranging from 0.2 to 2.5 mM for dAMP and from 0.01 to 1.5 mM for Ara-AMP. All activity assays were performed in 50 mM HEPES buffer at pH 7.5, supplemented with 1 mM MgCl_2_, substrate, and 0.2 mg/mL PenF. Reactions were initiated by enzyme addition and terminated at different time points. Product formation was measured to calculate reaction rates. Kinetic parameters were derived from the analysis of measured activities using the Lineweaver-Burk plot.

### 2.10. Sequence Alignment and Molecule Simulation

The amino acid sequence of PenF was obtained from the NCBI database. The three-dimensional structure of PenF was predicted using AlphaFold2 (https://colab.research.google.com/github/sokrypton/ColabFold/blob/main/AlphaFold2.ipynb, accessed on 6 November 2023) [20]. Sequences and structural data for the metal-dependent nucleotide pyrophosphohydrolase YpgQ (5DQW) and the uncharacterized protein YedJ (3GW7) were retrieved from the PDB database. Multiple sequence alignment was performed, and the secondary structure of PenF was visualized using the ESPript 3.0 (https://espript.ibcp.fr/ESPript/ESPript, accessed on 4 January 2024) web-based server [21].

Molecule docking of substrates into the active site of PenF was carried out using molecular operating environment (MOE) software (CCG, Montreal, Canada). Substrate structures were constructed in ChemDraw Ultra 8.0. All substrates were quickly prepared and docked into the active site of PenF. All docking experiments were performed at least five times. The final structure and locations were selected based on favorable root mean square deviation (RMSD) values and binding energy. Structural representations were generated using PyMOL.

### 2.11. Phylogenetic Tree

Amino acid sequences of 69 microbial 5′-nucleotidases were retrieved from the NCBI GenBank database. Sequence alignment, phylogenetic tree construction, and visualization were conducted using MEGA version 7.0.

## 3. Results and Discussion

### 3.1. Sequence and Phylogeny Analysis of PenF

Conserved domain analysis of the PenF peptide sequence revealed that it belongs to the HD-domain protein superfamily, indicating that it is likely a metal-dependent hydrolase. Comparative sequence analysis demonstrated significant amino acid sequence similarity between PenF and other HD-domain proteins. Specifically, PenF shares 57.1% similarity with an HD superfamily phosphodiesterase from *Nonomuraea Jabiensis* (A0A7W9GE97).

HD-domain hydrolases are divided into two enzymatic classes: the phosphatase subfamily and the phosphodiesterase subfamily. The phosphatase subfamily is further subdivided into mono-, di-, and triphosphohydrolases based on their substrates [22]. Given the function of the gene cluster and the loss of Ara-A synthesis upon deletion of *penF* in *Streptomyces antibioticus* NRRL 3238 [23], it is hypothesized that PenF functions as a 5′-nucleotidase-like monophosphatase.

The phylogenetic analysis (Figure 1) places PenF in a distinct position, showing a distant relationship with other known 5′-nucleotidases. This suggests that PenF represents a unique class of 5′-nucleotidases. However, further investigation is warranted to fully understand the specific properties and catalytic functions of PenF.

### 3.2. Characterization and Activity Assay of PenF

Recombinant PenF was successfully overexpressed in *E. coli* BL21 (DE3) and purified to homogeneity. SDS-PAGE analysis of recombinant PenF revealed a single band with an approximate molecular mass of 45 kDa (Appendix A). The mass of PenF was examined with an approximate molecular mass of 42 kDa using MALDI-TOF (Appendix A), consistent with the theoretical prediction.

5′-Nucleotidases catalyze the hydrolysis of a multitude of (deoxy)nucleotide-based substrates into their corresponding nucleosides and phosphate. The in vitro 5′-nucleotidase activity of PenF was tested using dAMP as the substrate. As shown in Figure 2A,B, PenF successfully catalyzed the hydrolysis of dAMP into deoxyadenosine, confirming its 5′-nucleotidase activity.

The enzymatic activity of PenF was evaluated in various buffer systems across a pH range of 2.0–11.0 using dAMP as the substrate. As depicted shown in Figure 2C, PenF exhibited optimal activity at pH 6.0-pH 7.5, with the highest enzymatic activities measured at 19.7 μmol/min/mg in phosphate buffer (pH 6.0) and 20.2 μmol/min/mg in HEPES buffer (pH 7.5). Interestingly, no activity was detected in the citrate-NaOH buffer at pH 6.0. At pH 7.5, the enzyme activity in phosphate and Tris-HCl buffers decreased to 7.6 μmol/min/mg and 0.95 μmol/min/mg, respectively, representing 37.4% and 4.7% of the activity in HEPES buffer. Furthermore, PenF activity was abolished under both highly acidic and alkaline pH conditions. These findings indicate that PenF is highly sensitive to both pH and buffer composition. The observed optimal pH values of PenF are similar to those reported for other 5′-nucleotidases, such as NE5-10 and Y2F8-2 from *Streptomyces* [24], which also exhibits peak activities at pH 7.5 and pH 6.0. The near-neutral pH optimum supports the notion that PenF functions as an intracellular enzyme [25].

### 3.3. Effect of Chemical Agents on Enzyme Activity

HD-domain proteins universally possess an HD residue dyad that coordinates transition metal ions such as Mn^2+^, Mg^2+^, Co^2+^, Fe^2+^, Zn^2+^, Cu^2+^, and Ni^2+^ [22,26]. The activity of HD-domain phosphatases varies based on their metal ions–cofactor dependencies [22]. To explore the metal ion dependency of PenF, we assessed its activity using dAMP as a substrate in the presence of 1 mM concentrations of various metal ions.

As shown in Figure 2D, both Mg^2+^ and Mn^2+^ significantly enhanced PenF activity, increasing it approximately 8.5-fold compared to the control (without adding any metals or chelators). Additionally, Co^2+^, Ni^2+^, and Fe^2+^ elevated enzyme activity to varying degrees, while Ca^2+^ had no effect. In contrast, PenF activity was completely inhibited by Zn^2+^, Cu^2+^, and EDTA. Notably, when an equivalent amount of Mg^2+^ was introduced to the reaction containing EDTA, PenF activity was recuperated, reaching double that of the control. These results confirm that PenF is an Mg^2+^/Mn^2+^-dependent nucleotide-5′-monophosphatases. This observation is particularly interesting, as previous studies have commonly shown Co^2+^ or Mn^2+^ to be the most effective cofactors for monophosphatases, whereas Mg^2+^ or Mn^2+^ are typically associated with triphosphatases [22]. The unique preference of PenF for Mg^2^⁺ and Mn^2^⁺ highlights its distinct enzymatic properties.

### 3.4. Substrate Specificity and Catalytic Activity of PenF

Characterized 5′-nucleotidases typically exhibit broad substrate specificity, though they often show a preference for certain substrates. For example, YfbR from *E. coli* and YGK1 from *Saccharomyces cerevisiae* demonstrate high phosphatase activity towards deoxyribonucleoside 5ʹ-monophosphates, particularly favoring 5ʹ-dGMP [27,28]. To assess PenF’s substrate specificity, we tested its activity against various nucleoside 5′-monophosphate, deoxyriboside 5′-monophosphate, and 2′(3′)-AMP. Given that *penF* is located in a gene cluster involved in the synthesis of arabinofuranosyladenine (Ara-A, vidarabine), PenF’s activity towards Ara-AMP was also evaluated.

As illustrated in Figure 3A, PenF displayed catalytic activity of 2.6 μmol/min/mg and 1.1 μmol/min/mg towards XMP and AMP, respectively, but minimal activity towards other nucleoside 5′-monophosphates and 2′(3′)-AMP. Among deoxyriboside 5′-monophosphates, PenF exhibited the highest activity against dAMP (14.5 μmol/min/mg), while showing lower activity against dGMP (1.3 μmol/min/mg) and dIMP (1.0 μmol/min/mg). Notably, PenF catalyzed the hydrolysis of Ara-AMP into Ara-A (Figure 3B,C). PenF demonstrated remarkably high catalytic activity towards Ara-AMP, with a rate of 32.7 μmol/min/mg, which is 2.2-fold higher than its activity against dAMP.

Given these findings, the kinetic parameters of PenF were investigated using Ara-AMP or dAMP as substrates (Table 2, Figure 3D,E). For dAMP, the values of *V*_max_, *k_cat_*, *K_m_*, and *k_cat_*/*K_m_* were calculated as 11.9 μM/min, 2.4 min^−1^, 344 μM, and 7.0 mM^−1^ min^−1^, respectively. For Ara-AMP, the respective values were 169.5 μM/min, 33.9 min^−1^, 71.5 μM, and 474.1 mM^−1^ min^−1^, respectively. The lower *K*_m_ and higher *k*_cat_ values for Ara-AMP suggest that PenF has a clear preference for Ara-AMP, corroborating its higher catalytic activity observed for this substrate.

These results highlight PenF as a metal-dependent nucleoside 5ʹ-monophosphatases activity with a unique preference for arabinose nucleoside 5ʹ-monophosphate over 2ʹ-deoxyribonucleoside 5ʹ-monophosphate and ribonucleoside 5ʹ-monophosphate. Notably, PenF is the first identified 5′-nucleotidase exhibiting specificity for arabinoside 5′-monophosphates.

### 3.5. PenF Structural Analysis

The crystal structure of PenF was predicted using AlphaFold2, revealing a core substrate binding architecture akin to the diphosphatase YpgQ from *Bacillus subtilis* (PDB 5DQW) (Figure 4A). Conserved residues responsible for binding metal ions and substrates were identified within PenF (Figure 4B, Appendix A). The residues His27, His56, Asp57, and Asp128 form a conserved “H…HD…D” motif crucial for metal ion coordination within PenF.

To further understand the specific activity, docking studies of PenF with Ara-AMP and dAMP were conducted to investigate substrate interactions. The docking results were assessed based on docker interaction energy and a favorable RMSD of ligands. The docking interaction energies were −67.34 kcal/mol for Ara-AMP and −61.72 kcal/mol for dAMP. PenF exhibited a higher theoretically affinity for Ara-AMP compared to dAMP, consistent with the observed enzymatic kinetics.

In the PenF-dAMP/Ara-AMP docking models, substrates were positioned near the metal ions binding site, with their phosphate ends oriented toward the metal ion (Figure 5A). Interactions between the substrates and PenF were characterized by ligand-protein interactions involving predicted catalytic residues near the active site (Figure 5B,C). In the PenF-dAMP complex, the adenine group predominantly interacted with His174 through π-π stacking. In contrast, the adenine group of Ara-AMP interacted with both Phe144 and His174 through π-π stacking. Additionally, Glu150 formed a hydrogen bond with the amino group of the adenine at distances of 3.2 Å for dAMP and 3.3 Å for Ara-AMP, stabilizing the substrate binding. The phosphate groups of dAMP and Ara-AMP primarily interacted with His23, Arg140, and Lys178. The hydrogen bond distances between His23 or Arg140 and the phosphate group were similar for both dAMP and Ara-AMP. However, the hydrogen bond distance between Lys178 and the phosphate group was shortened from 5.5 Å for dAMP to 3.9 Å for Ara-AMP, enhancing the interaction with Ara-AMP. A key difference was observed in sugar interactions: no residues located in the active site interacted with the deoxyribose group of dAMP, whereas Lys178 formed a hydrogen bond with the 2′ hydroxyl group of the arabinose group of Ara-AMP at a distance of 3.4 Å. This finding underscores PenF’s stronger interaction with Ara-AMP and provides molecular insights into its substrate preference.

To validate the role of key residues in substrate binding and catalysis, alanine scanning mutagenesis was performed on four predicted critical residues: Arg140, Phe144, His174, and Lys178. Four site-direct mutants (R140A, F144A, H174A, and K178A) were constructed, heterologously expressed, extracted and purified (Appendix A). The 5′-nucleotidase activities of these mutants were compared with WT PenF using dAMP and Ara-AMP as substrates. WT PenF exhibited catalytic activities of 48.5 ± 0.7 μmol/min/mg and 10.4 ± 0.3 μmol/min/mg for Ara-AMP and dAMP, respectively. As shown in Figure 6, mutation H174A had a minimal impact on activity against dAMP, indicating that the π-π interaction between His174 and adenine is not critical for substrate binding. In contrast, K178A, R140A, and F144A mutants showed reduced activity to 29.6%, 12.3%, and 64.9% of WT PenF activity against dAMP, respectively. All mutants showed significantly reduced activity against Ara-AMP, with R140A showing a nearly complete loss of activity. The activities of F144A, H174A, and K178A mutants against Ara-AMP were 6.5%, 17.9%, and 14.3% of WT PenF activity, respectively. These results highlight the essential roles of Arg140, Phe144, His174, and Lys178 in the PenF binding and catalysis of Ara-AMP, with Arg140 and Phe144 being particularly crucial.

We compared the amino acid residues within the ribose binding site of PenF, YpgQ, and YfbR. As shown in Appendix A, the amino acid residues adjacent to the 2′-C position of ribose in YpgQ and YfbR are hydrophobic, which does not favor the recognition and binding of hydroxyl groups. In PenF, however, the presence of basic amino acid residues aligned with the 2′-hydroxyl group on the same side of the sugar ring facilitates the enzyme’s recognition of arabinosyl substrates. Based on observed changes in the dAMP and ara-AMP activity of the K178A mutant and the intermolecular interactions involved, we propose that Lys178 primarily supports recognition and binding of the 2′-hydroxyl group in ribose or arabinose. Mutant activity assays against AMP indicate that K178A’s activity was 31% of the wild-type enzyme’s activity toward AMP, providing further evidence for this hypothesis.

## 4. Conclusions

In conclusion, we have successfully purified and characterized PenF, an arabinoside monophosphate specific 5′-nucleotidase-like enzyme PenF from *Streptomyces antibioticus*. Sequence and phylogeny analysis suggested that PenF represents a novel class of 5′-nucleotidases. Comprehensive biochemical analysis revealed that recombinant PenF is Mg^2+^/Mn^2+^-dependent and exhibits optimal 5′-nucleotidase activity under specific buffer conditions and pH values. PenF displays nucleoside 5ʹ-monophosphatase activity with a marked preference for Ara-AMP over 2ʹ-deoxyribonucleoside 5ʹ-monophosphate and ribonucleoside 5ʹ-monophosphate. This specificity suggests a role for PenF in the dephosphorylation of precursors involved in Ara-A biosynthesis.

As an HD-domain protein, PenF features a conserved “H…HD…D” motif involving His27, His56, Asp57, and Asp128, which coordinates Mg^2+^/Mn^2+^ within the enzyme. Structural and functional studies reveal that PenF interacts more strongly with Ara-AMP than with dAMP. Key residues outside the major substrate-binding site play crucial roles in substrate preference and catalytic activity. Taken together, for the first time these findings identify PenF as the 5′-nucleotidase specifically targeting arabinoside 5ʹ-monophosphate. Given its unique substrate specificity, PenF holds potential as a tool for the in vivo biosynthesis of nucleoside arabinosides.

## Data Availability

Data will be made available on request from the corresponding author.

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
