# Peer review of "Biochemical Characterization of an Arabinoside Monophosphate Specific 5′-Nucleotidase-like Enzyme from Streptomyces antibioticus"

_biomolecules, 2024, doi:10.3390/biom14111368_

Round 1
Reviewer 1 Report
Comments and Suggestions for Authors
Article is interesting. The authors did excellent research. But it is necessary to correct some details.
1. In the scheme 1 authors could put dAMP reaction also. This is for major clarity in the reaction comprehension.
2. In methodology, there are sections avoid. For example, there is no phylogenic section, DNA sequencing (place?).
3. How did the authors cell lysis?
4. When authors use %, they do not indicate (W/W) o (W/V).
5. In figure 2B, continuous line in pH is not necessary.
6. In figure 2 B and C show the same. I suggest to eliminate figure 2C.
7. In table 2, Vmax must be in capital letter.
8. Line 347, mutation number is wrong, K187A.
9. Review the microorganism nomenclature.
Questions
1. In results, in the phylogenetic tree, authors did not why they chose that sequences? What do they want to show in phylogenetic tree? PenF is not a laccasse. Why authors (in 209 line) talk about a laccases? It is necessary to do a better discussion about phylogenetic tree and the PenF in enzyme activity evolution. Are sequences choosing representative of 5´nucleotidases?
2. In the EDTA experiment with PenF, could PenF recuperate enzymatic activity when Mg+2 was restabled?
3. In figure 3C, in the plot, there are no dates of enzymatic activity between 200-500 mM. Why?
4. Section 3.5. it is necessary a deeper discussion about theirs findings and to compare with other investigations. A table comparing with others 5´nucleotidases could better this section.
Author Response
Comment 1. In the scheme 1 authors could put dAMP reaction also. This is for major clarity in the reaction comprehension.
Response 1: Thank you for your valuable suggestion. We have included the dAMP reaction in Scheme 1 in the revised manuscript to enhance clarity.
Comment 2. In methodology, there are sections avoid. For example, there is no phylogenic section, DNA sequencing (place?).
Response 2: Thank you for pointing this out. We have added these sections in the revised manuscript.
Comment 3. How did the authors cell lysis?
Response 3: Thank you for your comment. In this study, cells were lysed by sonication. This detail has been added to the revised manuscript (Section 2.4: Preparation of recombinant protein).
Comment 4. When authors use %, they do not indicate (W/W) or (W/V).
Response 4: Thank you for highlighting this. The glycerol concentration percentage has now been specified in the revised manuscript.
Comment 5. In figure 2B, continuous line in pH is not necessary.
Response 5: Thank you for your suggestion. We have removed the continuous line in Figure 2B in the revised manuscript.
Comment 6. In figure 2 B and C show the same. I suggest to eliminate figure 2C.
Response 6: Thank you for your suggestion. We have removed the original Figure 2C from the revised manuscript.
Comment 7. In table 2, Vmax must be in capital letter.
Response 7: Thank you for noting this. We have corrected it in the revised manuscript.
Comment 8. Line 347, mutation number is wrong, K187A.
Response 8: Thank you for catching this error. It has been corrected in the revised manuscript.
Comment 9. Review the microorganism nomenclature.
Response 9: Thank you for your comment. We have reviewed and corrected all microbial nomenclature throughout the manuscript.
Question
- In results, in the phylogenetic tree, authors did not why they chose that sequences? What do they want to show in phylogenetic tree? PenF is not a laccasse. Why authors (in 209 line) talk about a laccases? It is necessary to do a better discussion about phylogenetic tree and the PenF in enzyme activity evolution. Are sequences choosing representative of 5´nucleotidases?
Response 1: Thank you for this question. We constructed the phylogenetic tree to highlight the potential uniqueness of PenF as a 5'-nucleotidase. For this, we retrieved 69 microbial 5'-nucleotidase sequences from the NCBI database to build the tree with PenF. The reference to "laccase" was an error. In the evolutionary tree, the sequences most closely related to PenF are the 5'-nucleotidases from E. coli and Bacillus subtilis.
- In the EDTA experiment with PenF, could PenF recuperate enzymatic activity when Mg2+ was restabled?
Response 2: Thank you for your question. When an equivalent amount of Mg²⁺ was added to the reaction containing EDTA, PenF activity was restored, reaching twice that of the control reaction without added metals or chelators.
- In figure 3C, in the plot, there are no dates of enzymatic activity between 200-500 mM. Why?
Response 3: Thank you for your question. The concentration of Ara-AMP used in our experiments ranged from 0.01 to 1.5 mM, which provided a broad enough range, so no intermediate points between 200-500 µM were included. The kinetic constants were recalculated, and the results remain consistent.
- Section 3.5. it is necessary a deeper discussion about their findings and to compare with other investigations. A table comparing with others 5´nucleotidases could better this section.
Response 4: Thank you for your suggestion. In the final paragraph of Section 3.5, we compared PenF with other phosphohydrolases regarding the ribose-binding site. The presence of basic amino acid residues in PenF aligned with the 2'-hydroxyl group on the same side of the sugar ring greatly enhances PenF’s ability to recognize arabinosyl substrates.
Reviewer 2 Report
Comments and Suggestions for Authors
The manuscript presented by Wang and coworkers describes the thorough characterization of PenF in Ara-A biosynthesis. This HD-domain phosphatase is unusual in terms of pH dependency and metal dependency. The specificity of PenF towards arabinosyl substrate is the most intriguing feature of PenF as many other nucleotidases favor ribosyl or deoxyribosyl substrates. Overall, this is a relatively straightforward manuscript, and I would recommend publications if the authors can address the following comments.
Major:
1. For the metal dependency assay, there is no control experiment without adding any metals nor chelator. This control should be set as a reference for all the other experiments. In lines 254-256, if you consider adding Mg as the reference for 100% relative activity, you cannot say adding Mg significantly enhances the activity.
2. The authors are suggested to compare PenF and YpgQ regarding of the ribose binding site. The aim is to investigate why PenF prefers arabinosyl substrate while many other phosphatases prefer ribosyl substrate and which amino acid residue is critical for this selectivity. I’m also curious if the K178A mutant demonstrates improved activity over WT against AMP substrate.
3. According to Figure 5A, It appears that Ara-AMP binds in the same position as dAMP. However, there appears to be significant movement of active site residues, for example, Lys178. The authors are suggested to show a few key residues in Figure 5A so that the readers can visualize these differences.
Minor:
Line 21, it should be vidarabine 5’-monophosphate. There is a missing prime.
Line 68, it should be PTN, please check and correct the abbreviation throughout the manuscript.
Line 71, delete one “producing”.
Figure 2D, there should not be a 2+ charge for EDTA.
Line 128, Tris-HCl, missing H.
Line 247, please add a reference to this speculation.
Author Response
Comment 1. For the metal dependency assay, there is no control experiment without adding any metals nor chelator. This control should be set as a reference for all the other experiments. In lines 254-256, if you consider adding Mg as the reference for 100% relative activity, you cannot say adding Mg significantly enhances the activity.
Response 1: Thank you for your comments. We have now included a control experiment without added metals or chelators, which serves as a reference for all subsequent experiments in the revised manuscript. The interpretation of the effects of metal ions on enzyme activity has been revised accordingly.
Comment 2. The authors are suggested to compare PenF and YpgQ regarding of the ribose binding site. The aim is to investigate why PenF prefers arabinosyl substrate while many other phosphatases prefer ribosyl substrate and which amino acid residue is critical for this selectivity. I’m also curious if the K178A mutant demonstrates improved activity over WT against AMP substrate.
Response 2: Thank you for your insightful comments. YpgQ, a dNTP diphosphatase from Bacillus subtilis, and YfbR, a dAMP monophosphatase from Escherichia coli, were compared with PenF in terms of the properties of amino acid residues within the ribose binding site. As illustrated in Figure S4 of the revised manuscript, the amino acid residues adjacent to the 2'-C of ribose in YpgQ and YfbR are hydrophobic, which hinders the recognition and binding of hydroxyl groups. In contrast, the presence of basic amino acid residues in PenF, along with the 2'-hydroxyl group positioned on the same side of the sugar ring, greatly facilitates PenF's recognition of arabinosyl substrates. Based on the observed changes in dAMP and Ara-AMP activity of the K178A mutant and intermolecular interactions, we speculate that Lys178 plays a key role in recognizing and binding the 2'-hydroxyl group of ribose or arabinose. Activity analysis of the K178A mutant against AMP indicated that its activity was 31% of that of the wild type. This finding partially supports our hypothesis.
Comment 3. According to Figure 5A, it appears that Ara-AMP binds in the same position as dAMP. However, there appears to be significant movement of active site residues, for example, Lys178. The authors are suggested to show a few key residues in Figure 5A so that the readers can visualize these differences.
Response 3: Thank you for your suggestion. We have added a few key residues to Figure 5A to enhance visualization of these differences.
Comment 4 Minor:
Line 21, it should be vidarabine 5’-monophosphate. There is a missing prime.
Line 68, it should be PTN, please check and correct the abbreviation throughout the manuscript.
Line 71, delete one “producing”.
Figure 2D, there should not be a 2+ charge for EDTA.
Line 128, Tris-HCl, missing H.
Line 247, please add a reference to this speculation.
Response 4: Thank you for pointing out these errors. They have been corrected in the revised manuscript.